# Oncological Frontiers in the Treatment of Malignant Pleural Mesothelioma

**DOI:** 10.3390/jcm10112290

**Published:** 2021-05-25

**Authors:** Emanuele Vita, Alessio Stefani, Mariantonietta Di Salvatore, Marco Chiappetta, Filippo Lococo, Stefano Margaritora, Giampaolo Tortora, Emilio Bria

**Affiliations:** 1Comprehensive Cancer Center, Fondazione Policlinico Universitario Agostino Gemelli IRCCS, 00168 Roma, Italy; ale.stef92@gmail.com (A.S.); mariantonietta.disalvatore@policlinicogemelli.it (M.D.S.); giampaolo.tortora@policlinicogemelli.it (G.T.); 2Medical Oncology, Dipartimento di Medicina e Chirurgia Traslazionale, Università Cattolica del Sacro Cuore, 00168 Roma, Italy; 3Thoracy Surgery, Dipartimento di Medicina e Chirurgia Traslazionale, Università Cattolica del Sacro Cuore, 00168 Roma, Italy; marco.chiappetta@policlinicogemelli.it (M.C.); filippo.lococo@policlinicogemelli.it (F.L.); stefano.margaritora@policlinicogemelli.it (S.M.)

**Keywords:** malignant pleural mesothelioma, immunotherapy, target therapy

## Abstract

Malignant pleural mesothelioma (MPM) is a rare malignancy characterized by very poor prognosis and lack of treatment options. Immunotherapy has rapidly emerged as an effective tool for MPM, particularly for tumors of non-epithelioid histology. At the same time, comprehensive genomic sequencing may open the way to new-generation targeted-drugs able to hit specific MPM molecular vulnerabilities. These innovations will possibly enrich, but also dramatically complicate, the elucidation of treatment algorithms. Multidisciplinary integration is urgently needed.

## 1. Introduction

Malignant pleural mesothelioma (MPM) is a deadly malignancy arising from mesothelial cells of the pleural surface, accounting for fewer than 1% of all cancers [1,2,3]. Asbestos exposure, usually occurring in the workplace, is the leading cause of MPM through the induction of chronic inflammation and macrophages releasing DNA-mutagenic oxidizing agents. Other risk factors include occupational radiation and prior chest radiotherapy [4]. Very rarely, germline mutations in breast-related cancer antigens (BRCA)-associated protein 1 (BAP1) can be passed in families [5,6].

The histological classification of MPM includes three main subtypes: epithelioid, sarcomatoid (including the desmoplastic and lymphohistiocytic variants), and biphasic. The epithelioid histology is associated with a more favorable prognosis and occurs in 60–80% of patients, whereas the sarcomatoid histology (20% of cases) has worse outcomes, with a lower chance of response to therapy [7].

Multimodality therapy including induction platinum-based chemotherapy, surgical resection (pleurectomy/decortication with mediastinal lymph node sampling or extrapleural pneumonectomy), and sometimes radiation therapy is generally offered to young patients with good performance status, localized disease, and epithelioid histological subtype [8,9].

Based on the results of the EMPHACIS trial, combination therapy with cisplatin (CDDP) and pemetrexed (PEM) has been for long the cornerstone of first-line treatment for patients with advanced, unresectable MPM [10]. The carboplatin–PEM regimen showed comparable efficacy to CDDP–PEM in a phase II study; therefore, in clinical practice it should be preferred for patients with a poor performance status (PS) and/or comorbidities [11]. The clinical role of second-line therapy for progressive or relapsed disease is still undefined, and no post-progression validated treatment has emerged. Pemetrexed-based re-treatment should be considered for patients who have obtained a PFS greater than 3 months with first-line therapy [12,13], while other active drugs, such as gemcitabine and vinorelbine, can be used for platinum-refractory patients with a good PS [14,15,16].

However, in a comprehensive perspective, the introduction of the pemetrexed-based strategies has produced negligible survival improvements, and the prognosis of MPM is still very poor with an overall 5-year survival rate < 10%, further underscoring the urgent need for more effective therapies. In the last few years, new therapeutic approaches focusing on three different research areas (immunotherapy, functional loss of tumor suppressor genes, and angiogenesis) have been investigated for MPM treatment. Here, we critically review these new emerging options of treatment for MPM, moving from the actual therapeutic strategies to upcoming practice-change future approaches.

## 2. Immunotherapy

Immunotherapy (IO) has opened a new era in the management of thoracic malignancies, and several immune checkpoint inhibitors, targeting the cytotoxic lymphocyte antigen 4 (CTLA4) and programmed Death-1/Programmed Death-Ligand 1 (PD-1/PD-L1) signaling axis, have been approved for the treatment of lung cancers.

In the last years, several clinical trials have successfully investigated the activity of IO in MPM treatment, firstly for recurrent/relapsed disease and, more recently, as an upfront treatment compared to platinum–pemetrexed-based chemotherapy (Table 1).

In a retrospective analysis conducted by Patil and colleagues [17], a sample of 99 MPM specimens were profiled for immune gene expression and PD-L1 expression, proposing a classification in three subgroups according to the degree of inflamed phenotype: 60% of the samples analyzed showed an inflamed status, making mesothelioma a good theoretical candidate to immunotherapy.

**Table 1 jcm-10-02290-t001:** Selected Clinical Trials investigating Immunotherapy in MPM.

Name	Trial ID	IO Agent	Phase	No. pts	Treatment Arms	Result/ Status	Endpoint
**Relapsed/Recurrent MPM**							
MESOT-TREM-2008 [18]	NCT01649024.	Tremelimumab	II	25	Tremelimumab (15 mg/kg every 90 days)	Negative	ORR
MESOT-TREM-2008 [19]	NCT01655888.	Tremelimumab	II	29	Tremelimumab (10 mg/kg every 4 weeks)	Negative	ORR
DETERMINE [20]	NCT01843374.	Tremelimumab	IIB	571	Temelimumab (10 mg/kg) vs. placebo	Negative	OS
KEYNOTE-028 [21]	NCT02054806	Pembrolizumab	I	25	Pembrolizumab (10 mg/kg q14)	/	ORR
KEYNOTE-158 [22]	NCT02628067	Pembrolizumab	II	118	Pembrolizumab 200 mg q21 up to 35 cycles	Negative	ORR
PROMISE-Meso [23]	NCT02991482	Pembrolizumab	III	114	Pembrolizumab vs. CHT	Negative	PFS
JAVELIN Solid Tumor [24]	NCT01772004	Avelumab	IB	53	Avelumab (10 mg/kg q14)	Negative	ORR
NivoMes [25]	NCT02497508	Nivolumab	II	38	Nivolumab (3 mg/kg q14)	Positive	DCR
MERIT [26]	JapicCTI163247	Nivolumab	II	34	Nivolumab (3 mg/kg q14)	Positive	ORR
CONFIRM [27]	NCT03048474	Nivolumab	III	332	Nivolumab (240 mg q14)	Positive	PFS/OS
NCT03075527 [28]	NCT03075527	Tremelimumab + Durvalumab	II	19	Trem + Durv (4 Cycles) − Durv	Negative	ORR
NIBIT-Meso-1 [29]	NCT02588131	Tremelimumab + Durvalumab	II	40	Trem + Durv (4 Cycles) − Durv	Positive	ORR
MAPS2/IFCT1501 [30]	NCT02716272	Ipilimumab + Nivolumab	II	125	Nivolumab +/− Ipilimumab	Positive	12W DCR
INITIATE [31]	NCT03048474	Ipilimumab + Nivolumab	II	35	Nivolumab + Ipilimumab	Positive	12W DCR
**Upfront treatment**							
Checkmate 743 [32]	NCT02899299	Ipilimumab + Nivolumab	III	92	CDDP + PEM vs. IPI + NIVO	Positive	OS
IND-227	NCT02784171	Pembrolizumab	II-III	520	CDDP + PEM +/− PEMBRO	Active, not recruiting	PFS/OS
PrE505 [33]	NCT02899195	Durvalumab	II	55	CDDP + PEM + DURVA	Positive	OS
DREAM [34]	ACTRN 12616001170415	Durvalumab	II	54	CDDP + PEM + DURVA	Positive	PFS
DREAM3R	NCT04334759	Durvalumab	III	480	CDDP + PEM +/− DURVA	Recruiting	OS
ETOP BEAT-meso trial	NCT03762018	Atezolizumab	III	320	CBDCA + PEM + BEVA +/− ATEZO	Recruiting	PFS, OS

List of abbreviations: Trem = Tremelimumab; Durva: Durvalumab; IPI = ipilimumab; NIVO = Nivoliumab; PEMBRO = pembrolizumab; CDDP = cisplatino; PEM = pemetrexed; ATEZO = atezolizumab; ORR = objective response rate; DCR = disease control rate; 12W DCR = disease control rate at 12 weeks; PFS = progression-free survival; OS = overall survival.

### 2.1. Single-Agent Immunotherapy

To date, tremelimumab is the only an anti-CTLA4 inhibitor tested as monotherapy in MPM. Based on encouraging clinical and immunological activity in the two single-arm MESO-TREM studies [18,19], tremelimuab was tested in a larger placebo-controlled trial. In the DETERMINE study [20], 571 pre-treated MPM patients were randomized (2:1) to tremelimumab (10 mg/kg every 4 weeks for seven cycles and then every 12 weeks) or placebo. There were no significant differences in response or survival between the two groups (mOS 7.7 months for tremelimumab vs. 7.3 months for placebo (*p* = 0.408). Although there seemed to be a trend in the sarcomatoid group in favor of tremelimumab, the number of patients was too small to detect a significant difference.

In the phase 1b trial KEYNOTE-028, which evaluated pembrolizumab (an anti-PD-1 mAb) 10 mg/kg q14 in PD-L1-positive solid tumors, a cohort of 25 patients with MPM exhibited a median OS of 18 months and DCR of 72%, with 4 patients maintaining a response for about two years [21]. In the multicohort, single-arm, phase 2 KEYNOTE-158 study [22], 118 patients with pre-treated MPM and biomarker-evaluable tumor samples were enrolled to receive pembrolizumab 200 mg intravenously every 3 weeks for up to 35 cycles. The primary study endpoint was ORR, and only 10 patients (8%; 95% CI; 4–15) had an objective response; the median DOR was 14·3 months (range: 4.0 to over 33.9), and 60% of objective responses were ongoing at 12 months. The median overall survival was 10·0 months (95% CI 7.6–13.4), and the median progression-free survival was 2.1 months (2.1–3.9). Objective responses were observed independently of PD-L1 expression (6/77 PD-L1+ MPM; mDOR 17.7 months [range 5.8–33.9+] and 4/31 PD-L1-negative MPM; mDOR 10.2 months [4.0–16.6]). Similarly, in the phase 3 PROMISE-Meso trial [23], 114 patients with pre-treated MPM (notably, almost 90% of patients had an epithelioid histology) were randomized to receive Pembrolizumab or investigator’s choice chemotherapy (gemcitabine or vinorelbine). Despite an ORR of 22% for pembrolizumab (vs. 6% in the chemotherapy arm), mOS was 10.7 months in the experimental arm versus 11.7 months in the control arm.

Avelumab (anti-PD-L1 mAb) as a single agent in 53 pe-treated MPM was tested in the phase Ib JAVELIN Solid Tumor trial [24], achieving a dismal mOS of 10.7 months, although in patients achieving a response, the median DOR was 15.2 months.

In the NivoMes phase 2 study, a population of 38 patients with relapsed MPM was treated with nivolumab 3 mg/kg q14, obtaining a 3-month DCR of 50% and an ORR of 24%. The role of nivolumab as a salvage therapy was confirmed by the phase 2 MERIT trial and, most recently, by the phase 3 placebo-controlled CONFIRM trial: 332 patients were randomized 2:1 to nivolumab 240 mg q14 or placebo, stratified by histology (epithelioid vs. non epithelioid); the study met its two co-primary endpoints, showing an investigator-assessed mPFS of 3.0 vs. 1.8 months (HR 0.62, *p* < 0.001) and an investigator-assessed mOS of 9.2 vs. 6.6 months (HR 0.72, *p* = 0.02) in favor of nivolumab [25,26,27].

### 2.2. Combination Therapy

As seen in other malignancies, there can be an additive or synergic effect when combining ICIs or ICI with chemotherapy. Two phase 2 trials evaluated the activity of tremelimumab plus durvalumab in relapsed MPM. The NCT03075527 trial did not meet its endpoints of activity in the interim analysis [28]. The NIBIT-Meso-1 phase 2 trial enrolled 40 patients who were treated with tremelimumab (1 mg/kg) and durvalumab (20 mg/kg) every 4 weeks for four cycles, followed by maintenance with durvalumab up to nine cycles; the results were promising, as 28% of patients achieved a PR, and mOS was 16.6 months [29].

The phase 2 MAPS2/IFCT1501 trial was a two-arm non comparative study where 125 patients were randomized 1:1 to nivolumab (3 mg/kg every 2 weeks) or nivolumab plus ipilimumab (1 mg/kg every 6 weeks): the primary endpoint was a 3-month DCR > 40% that was reached in both arms (44.4% in the nivolumab arm and 50% in the nivolumab plus ipilimumab arm); ORR was 26% in the combination arm and 19% in the nivolumab arm, while mOS was, respectively, 15.9 months and 11.9 months [30]. The activity of this combo was confirmed in the phase 2 trial INITIATE, which evaluated 35 patients with MPM in the second-line setting treated with nivolumab 240 mg q14 plus ipilimumab 1 mg/kg every 6 weeks, achieving a DCR of 68% and an ORR of 29% [31].

With regard to the first-line scenario, the paradigm of treatment will be deeply transformed following the results of the phase 3 CheckMate 743. A total of 605 patients, stratified by histology (epithelioid vs. non epithelioid) and sex, were randomized 1:1 to standard platinum–pemetrexed chemotherapy or nivolumab (3 mg/kg q2w) plus ipilimumab (1 mg/kg q6w) until disease progression, inacceptable toxicity, or completion of two years of treatment. The study met its primary endpoint: mOS was 18.1 months in the experimental arm versus 14.1 months in the control arm (HR 0.74, *p* = 0.002). Nevertheless, considering the subgroup analysis, the benefit was not consistent in patients with an epithelioid histology, for whom mOS was 18.7 months versus 16.5 months (HR 0.86, 95%CI 0.69–1.08); on the contrary, the subgroup who showed the greatest survival advantage was the non-epithelioid one, which historically is refractory to the standard chemotherapy treatment: mOS was 18.1 months with the ICI combo versus 8.8 months in the control group (HR 0.46) [32].

This different response to immunotherapy is consistent with previous pre-clinical studies and may be related to the different tumor microenvironment of epithelioid and sarcomatoid/biphasic MPM. In the above-mentioned study [35] carried out by Pasello et al., the biphasic/sarcomatoid histotype was characterized by higher infiltration of CD8+ T lymphocytes and CD68+ macrophages and also by higher PD-L1 expression; these features, which are markers of aggressiveness, are associated with a lower response to chemotherapy but may be the reason why immunotherapy works better [17]. Similarly, Alay et al. analyzed a large collection (*n* = 516) of MPM and identified three subgroups according to the relative infiltration of cytotoxic T cells and T-helper 2 cells; the third group (high levels of cytotoxic T cells and low levels of T-helper2 cells) was characterized by an inflamed gene signature and by a better prognosis; the authors also speculated that this subgroup might show better response to immunotherapy [36].

The combination of PD-1-blocking agents and chemotherapy has been successfully evaluated in different solid malignancies, and multiple randomized studies are running also for MPM patients. The first results of combining durvalumab with cisplatin–pemetrexed in the first line are hopeful. In the Australian DREAM study [35], the primary endpoint was progression-free survival at 6 months (PFS6), measured according to mRECIST for MPM and analyzed in the intention-to-treat population: after a median 28.2-month follow-up, 31 (57%; 95% CI 44–70) of 54 patients were alive and progression-free at 6 months. Similarly, the American counterpart study (PrE0505) reported a median OS of 20.4 months, exceeding the pre-specified criteria for clinically meaningful improvement of 19.0 months, which corresponded to a 58% improvement in the median OS of 12.0 months associated with a pemetrexed–cisplatin historical control. The 6-, 12-, and 24-month OS rates were 87.2%, 70.4%, and 44.2%, respectively, while the corresponding PFS rates were 69.1%, 16.4%, and 10.9%. The median PFS was 6.7 months [36]. An international world-wide phase III randomized study (DR3AM) with this combination is currently ongoing, and results are expected in 2024 (Clinicaltrials.gov no. NCT04334759).

The IND-227 study (Clinicaltrials.gov no. NCT02784171) has been initiated to determine the value of pembrolizumab in the first-line setting. The phase II part of this study had three treatment arms: single-agent pembrolizumab, cisplatin/pemetrexed, or a combination of the three agents. In the ongoing phase III part, the patients are randomized to platinum–pemetrexed plus pembrolizumab or to the same chemotherapy alone. Primary results will be available in 2022.

The ETOP BEAT-meso trial (Clinicaltrials.gov no. NCT03762018) is currently enrolling, and 320 patients will be randomized so to receive platinum–pemetrexed–bevacizumab with or without atezolizumab. The primary endpoint is PFS. The first results are expected in 2024.

## 3. Targeting Functional Loss of Tumor Suppressor Genes (TSGs)

In an effort to identify actionable targets in MPM, the use of massive parallel sequencing has revealed frequent deletions or loss-of-function mutations of tumor suppressor genes (TSGs), most often cyclin-dependent kinase inhibitor 2A (CDKN2A), BRCA1-associated protein-1 (BAP1), and chaperone proteins [37,38]. Despite TSG are not directly targetable, aberrant cancer genome rewires biochemical networks, leading to synthetic lethal vulnerabilities and providing alternative approaches for targeting TSG-driven MPM (Table 2).

### 3.1. CDKN2A

The cyclin-dependent kinase inhibitor 2A (CDKN2A) is a tumor suppressor gene located at chromosome 9p21.3 that encodes two functionally unrelated proteins, i.e., p16INK4a and p14ARF. The p16INK4a protein is a CDK inhibitor that acts in the inactivation of retinoblastoma proteins (Rb), leading to failure of cell cycle arrest. The p14ARF protein is a key protein cell cycle regulator that inhibits the degradation of p53 [38,39,40]. Loss of the 9p21 locus is common in MPM, and CDKN2A deficiency is potentially associated with vulnerability to CDK4/6-targeted therapies. In the SIGNATURE trial, the efficacy of Ribociclib in CDK4/6 pathway-activated malignancies including five MPM was tested, with a dismal ORR of 2.9% (Clinicaltrials.gov no. NCT02187783). Abemaciclib is being investigated in MPM bearing p16Ink4a deficiency, as an arm (MiST2) of a larger molecular-driven phase II trial (Clinicaltrials.gov no. NCT03654833).

### 3.2. BAP-1

The BRCA1-associated protein 1 carboxy-terminal hydrolase (BAP1) is a tumor suppressor gene that encodes a deubiquitinating enzyme that plays a crucial role in the regulation of several biological processes, including DNA double-strand breaks (DSBs) response and epigenetic regulation through chromatin remodeling. Germline mutations in BAP1 have been identified in families with “BAP1 cancer syndrome”, characterized by the predisposition to developing benign atypical melanocytic lesions, uveal melanomas, and MPM. Additionally, BAP1 somatic mutations/inactivation have been also frequently found in sporadic epithelioid MPM (57–63%) and have been associated with a better response to platinum-based chemotherapy [36,37,41,42]. Similar to BRCA1/2-deficient cancers, mutation in the BAP1 gene leads to homologous recombination-deficient (HRD) tumors and increases the reliance on poly ADP ribose polymerase (PARP)-mediated DNA repair pathways; therefore, PARP1/2 inhibitors can induce synthetic lethality in MPM.

In a single-arm, phase II trial with prospective molecular stratification (Mesothelioma-Stratified Therapy 1 [MiST1]), patients with relapsed cytoplasmic-BAP1-deficient or BRCA1-deficient mesothelioma (pleural or peritoneal or other primary localization), received rucaparib 600 mg twice a day orally, for up to six cycles of 28 days. In this molecularly selected population, rucaparib met the primary outcome of the study, achieving 58% of disease control rate at 12 weeks (95% CI 37–77; 15/26 patients) and 23% at 24 weeks (95% CI: 9–44; 6/26 patients); all reported toxicities were manageable [43]. Niraparib, another PARP inhibitor, has also been evaluated in patients with BAP-1-negative metastatic relapsed or refractory solid tumors (ClinicalTrials.gov no. NCT03207347).

However, recent pre-clinical studies [44,45] showed that the BAP1 status does not determine the sensitivity to PARP inhibitors in patient-derived mesothelioma cell lines, which is surprising and in contrast with previous observations. Several possibilities can be envisioned to explain these discrepancies, including the presence of co-occurring mutations leading to a BAP-1-status-independent HRD phenotype and/or different BAP1 splice isoforms affecting the sensitivity of MPM cells to PARP inhibition. Consequently, further investigations about HRD status are needed to guide PARP-targeted therapy for patients with BAP1-mutant MPM.

Additionally to direct synthetic lethality, treatment of HRD-tumors with PARP inhibitors generates significant levels of DNA damage, which has the potential to further increase the tumor mutational burden, promoting neoantigen release and upregulating both interferons and PD-L1 expression, suggesting a potential complementary and synergistic role with immune checkpoint inhibitors. Based on this rationale, a phase II single-arm study has been planned to investigate efficacy and safety of the combination of niraparib and dostarlimab, a PD-1 inhibitor, in patients with HRD-positive and PD-L1 ≥ 1% advanced non-small-cell lung cancer (NSCLC) and/or MPM [46].

As a chromatin regulator, BAP1 works as the catalytic subunit of the Polycomb repressive deubiquitinase (PR–DUB) complex that removes mono-ubiquitin from histone H2A [47]. Consequently, BAP1-altered MPM cells are critically dependent on the activity of enhancer of zeste homolog 2 (EZH2), the functional enzymatic component of the Polycomb Repressive Complex 2, an alternative transcriptional complex involved in histone methylation.

Vorinostat, a histone deacetylase inhibitor (HDI), was compared to placebo in a large phase III trial (VANTAGE-014) in patients with advanced MPM who had previously failed one or two chemotherapy regimens. Despite a statistically but not clinically significant improvement of PFS from 6.1 to 6.3 weeks (HR 0.75, 95%CI 0.63–0.88; *p* = 0.001), the study failed its primary endpoint OS (30.7 vs. 27.1 weeks; HR 0.98 95%CI 0.83–1.17; *p* = 0.86). Belinostat, another histone deacetylase inhibitor, did not show any clinical activity as well [48,49].

The selective EZH2 inhibitor tazemetostat was recently evaluated in a multipart phase II study including patients affected by relapsed or refractory MPM with BAP1 inactivation. Tazemetostat met the primary endpoint with 47% of 12-week DCR (*n* = 35/74), despite the ORR per RECIST version 1.1 was only 3% (*n* = 2/74). Grade ≥3 treatment-emergent adverse events (TEAEs) occurred in ≤5% of patients, and there was no treatment discontinuation or death due to TEAEs. Based on these findings, tamezostat showed antitumor activity in BAP1-deficient MPM with well-tolerated toxicity, supporting further clinical exploration [50].

### 3.3. Molecular Chaperones

Chaperone proteins assist other proteins to reach properly conformational folding and aid the assembly or disassembly of macromolecular structures. By helping to stabilize partially unfolded proteins, chaperone proteins are essential to face the increased demand for protein transporting across membranes required for tumor growth, providing a potential target for anti-cancer drugs.

Hsp90 (heat shock protein 90) is a molecular chaperone that mediates the post-translational stabilization of critical oncogenic signaling molecules, via a repertoire of client proteins that include oncogenic kinases relevant to MPM such as AXL and MET [51]. Additionally, thymidylate synthase is an Hsp90 client and can be downregulated by inhibition of Hsp90, enhancing DNA damage induced by antifolates and platinum chemotherapy [52,53].

Ganetespib, a highly selective small-molecule Hsp90 inhibitor, was combined with upfront pemetrexed–platinum chemotherapy in the phase I/II MESO-02 trial [54]. Results from the dose-escalation phase showed that the combination was well tolerated and had promising antitumor activity. At the maximum tolerated dose of ganetespib (200 mg/m^2^), ORR was 56% (10/18 patients), DCR was 83% (15/18 patients), and median PFS was 6.3 months (95% CI, 5.0–10.0). One responder exhibited disease control beyond 50 months. In preclinical assays [55], acquisition of aneuploidy has been reported as a mechanism of resistance to Hsp90, and in the exploratory analysis global loss of heterozygosity was associated with shorter time to progression (HR 1.12; 95% CI, 1.02–1.24; *p* = 0.018). Nevertheless, this result must be interpreted cautiously because increasing genomic instability per se may be negatively prognostic, and the study was underpowered to detect any interaction between specific copy number alterations and sensitivity to Hsp90 inhibition.

## 4. Targeting Angiogenesis

Mesothelioma cells produce high amounts of endothelial growth factor (VEGF) and express VEGFR-1 and VEGFR-2 receptors; therefore, angiogenic proteins play a key role either as autocrine growth factors or as vascular permeability inducers of pleural effusion [56,57,58,59,60]. Consequently, there is a strong rationale for inhibiting angiogenesis in MPM in order to improve symptoms, reduce the number of invasive pleural procedures, and prolong patient life.

The addition of anti-angiogenic agents to first-line platinum doublets has been investigated in two large phase III trials (Table 2). In the MAPS study, the addition of bevacizumab to CDDP–PEM only slightly prolonged median OS (mOS) in comparison to CDDP–PEM (18.8 vs. 16.1 months), at the cost of increased grade 3–4 toxicities and class-specific adverse events [61]. Despite National Comprehensive Cancer Network guidelines include the optional addition of bevacizumab to CDDP–PEM chemotherapy, this regimen is not licensed by Regulatory Agencies.

More recently, in the phase II SWOG S0905 trial [62], chemotherapy-naïve patients with MPM of any histologic subtype were randomly treated with cediranib or placebo and platinum–pemetrexed for six cycles, followed by maintenance cediranib or placebo. Adding cediranib did not produce improvement in the primary endpoint PFS (7.2 months vs. 5.6 months, HR = 0.71; 80% CI: 0.54 to 0.95; *p* = 0.062), and no difference in overall survival was observed (10 vs. 8.5 months, HR = 0.88 80% CI: 0.67–1.17; *p* = 0.28). Additionally, the cediranib arm reported more grade ≥3 toxicity, including diarrhea, dehydration, hypertension.

Similarly, in the phase III part of LUME-Meso trial, the combination of CDDP–PEM with nintedanib, an anti-angiogenic multikinase inhibitor, failed to improve PFS in chemotherapy-naïve patients with unresectable epithelioid MPM, despite earlier positive findings in the phase II part of the study [63].

In second-line setting, several multi-kinase inhibitors with antiangiogenic activity were tested in early-phase trials, but none of them showed convincing efficacy for continuing their clinical development [64,65,66].

NGR-hTNF is a vascular-targeting drug that increases the penetration of intratumoral chemotherapy and T cell infiltration by modifying the tumor microenvironment. The phase III NGR015 trial, including 400 patients with refractory MPM, compared the combination of NGR–hTNF and single-agent chemotherapy (gemcitabine, vinorelbine, or doxorubicin) with chemotherapy alone [67]. The study did not meet its primary endpoint, as overall survival did not differ between the two treatment groups (median 8.5 months [95% CI: 7.2–9.9] in the NGR–hTNF group vs. 8.0 months [6.6–8.9] in the placebo group; hazard ratio 0·94, 95% CI 0.75–1.18; *p* = 0.58). Nevertheless, patients with short treatment-free interval (TFI) after first-line therapy (<median 4.8 months) had better OS and PFS with the addition of NGR–hTNF to chemotherapy. Interestingly, this survival benefit was maintained after a 3-year follow-up, deserving a confirmatory randomized trial including only short TFI patients.

The RAMES Study, an Italian multicenter phase II trial, explored the efficacy and safety of the addition of ramucirumab (RAM), an antibody selectively directed against the extracellular domain of VEGFR-2, to gemcitabine (GEM) as a second-line treatment. The study showed a borderline statistically significant (HR 0.71; *p* = 0.057) and a clinically meaningful improvement of OS in the ramucirumab arm, with an increase of median value in the intention-to-treat population by more than 6 months (from 7.5 to 13.8 months). In the RAM–GEM arm, the survival advantage was not correlated to TTP at first-line therapy (13.6 months in TTP ≤ 6 months and 13.9 months in TTP > 6 months) and histotypes (13.8 months in the epithelioid and 13.0 months in non-epithelioid). The RAM–GEM combination showed a reasonable safety profile, with a low rate of severe adverse events, including class-related toxicities. The genetic profiling of tissue samples from 110/164 patients enrolled failed to discover predictive markers for response to ramucirumab [42,68].

**Table 2 jcm-10-02290-t002:** Selected Clinical Trials investigating targeted drugs in MPM.

	Trial ID	Target	Phase	No. pts	Result/Status	Endpoint	Biomarker
**Targeting functional loss of tumor suppressor genes (TSGs)**							
**CDKN2A**							
**Ribociclib**	NCT02187783	CDK4/6	II	106 (5 MPM)	Negative	ORR	CDK4/CDK6, CDKN2A CCND1/CCND3
**Abemaciclib**	NCT03654833	CDK4/6	II	120	Recruiting	DCR	P16INK4A
**BAP-1**							
Rucaparib [43]	NCT03412097	PARP 1/2	IIA	26	Positive	DCR	BAP1/BRCA1
Niraparib	NCT03207347	PARP 1/2	II	47	Recruiting	ORR	BAP-1/HRD
Vorinostat [48]	NCT00128102	HDAC	III	661	Negative	OS	None
Niraparib/dostarlimab [46]	NA	PARP 1-2/PD-1	II	35	Planned	PFS	HRD/PD-L1 ≥ 1%
Tazemetostat [50]	NCT02860286	EZH2	II	74	Positive	DCR	BAP-1
**Chaperones**							
Ganetespib [54]	NCT01590160	Hsp90	I-II	18	Positive	Safety, PFS	None
**Targeting angiogenesis**							
Bevacizumab [61]	NCT00651456	VEGF	III	448	Positive	OS	None
Cediranib [62]	NCT01064648	VEGFR/PDGFR	II	92	Negative	PFS	None
Nintedanib [63]	NCT01907100	VEGFR/PDGFR	III	458	Negative	PFS	None
NGR-hTNF [67]	NCT01098266	Multiple	III	400	Negative	OS	None
Ramucirumab [69]	NCT03560973	VEGFR	III	161	Positive	OS	None

List of abbreviations: CDK4/6: Cyclin-dependent kinase 4/6; CDKN2A: cyclin-dependent kinase inhibitor 2A; CCND1/CCND3: cyclin D1/D3; p16INK4a: cyclin-dependent kinase inhibitor 2A, CDKN2A, multiple tumor suppressor 1; PARP 1/2: poly [ADP-ribose] polymerase 1/2; BAP1: BRCA1-associated protein-1; BRCA: Breast cancer type 1 susceptibility protein; HRD: Homologous recombination Deficiency; HDAC: Histone deacetylases; PD(L)-1 Programmed cell death protein (ligand)-1; EZH2: enhancer of zeste homolog 2; Hsp90: heat shock protein 90; VEGF(R): Vascular endothelial growth factor (receptor); PDGFR: Platelet-derived growth factor receptors; ORR = objective response rate; DCR = disease control rate; PFS = progression-free survival; OS = overall survival.

## 5. Critical Discussion

MPM is a rare, aggressive malignancy with limited treatment options. The role of surgery is limited, because only few patients are surgical candidates and a complete microscopic resection is rarely realistic; moreover, the evidence supporting a multimodal strategy is weak due the lack of prospective randomized trials.

After decades of trial failures, immunotherapy is rapidly emerging as an important tool for the treatment of MPM, resulting in a fast change of treatment algorithms. In the pivotal phase III trial Checkmate 743, ipilimumab and nivolumab led to a significantly 4-month mOS improvement in a first-line setting compared to platinum-based doublet chemotherapy (18.1 vs. 14.1 months; HR 0.74 96.6% CI: 0.60–0.91; *p* = 0.0020). Nevertheless, the survival benefit in overall population seems to be mostly related to the overwhelming superiority of the IO combination in the non-epithelioid subgroup, where chemotherapy performed poorly, as expected (18.1 vs. 8.8 months; HR 0.46, 95% CI: 0.31–0.68). Conversely, the IO combination did not show clear superiority compared to chemotherapy in epithelioid MPMs (18.7 months vs. 16.5 months; HR: 0.86, 95%CI: 0.69–1.08). Despite the trial was not powered to detect differences among epithelioid and non-epithelioid subgroups, these findings cannot be ignored because the histological subtype was a stratification factor and the non-epithelioid histology subgroup was numerically sufficient (25% of the overall population) to affect overall outcomes. Additionally, the NIVO plus IPI treatment arm showed similar outcomes compared to the standard CT arm for mPFS (6.8 vs. 7.2 months, HR 1.00, 95% 0.82–1.2) and ORR (40% vs. 43%), but the responses were more durable in IO-treated patients (2-year DOR: 32% vs. 8%), suggesting the existence of undefined predictive biomarkers.

At the same time of immunotherapy development, comprehensive genomic studies have revealed cancer collateral vulnerabilities contextually related to specific genetic alterations, which offer promising molecularly driven strategies. Particularly, targeting BAP-1-deficient tumors through PARP inhibition and/or enhancing platinum–pemetrexed-associated lethality thought Hsp90 inhibitors resulted as effective translational opportunities, deserving further clinical development and integration in multimodality approaches. Lastly, anti-angiogenetic drugs failed to produce significant results in unselected patients, and the question rises whether angiogenesis should still be considered an adequate therapeutic target in MPM or whether identification and careful selection of patients on the basis of their pro-angiogenic tumor features is needed.

In conclusion, the above-mentioned innovative therapeutical strategies are likely to enrich, but also dramatically complicate, clinicians’ decision. Upfront immunotherapy marks a significant milestone for non-epithelioid MPM, which is usually refractory to chemotherapy and characterized by poorer prognosis. Nevertheless, a treatment algorithm for epithelioid MPM is far to be elucidated: in a future scenario, integration between new surgical techniques, genomic profiling, and emerging systemic therapies will be crucial for selecting the best treatment choices for each patient and maximize the survival benefits.

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
