# Peer review of "Oncological Frontiers in the Treatment of Malignant Pleural Mesothelioma"

_jcm, 2021, doi:10.3390/jcm10112290_

Round 1

Reviewer 1 Report

The authors wrote a concise and current review about approved systemic therapies as well as novel emerging options. The last mentioned options are explained in detail focussing on biochemical and molecular pathways and these options are critically discussed considering their chance to clinically and statistically significant improve the prognosis of the patients.

In the last years several reviews about systemic treatment options for patients suffering from malignant pleural mesothelioma were published. But well structurec tables displaying the relevant clinical studies and the accurate accompanying text haven´t yet been published. Furthermore, the review does`t contain too many phase II clinical studies which failed to meet their primary endpoints.

I have noticed that the abbreviation "HRD (homologous recombination deficient)" was distorted at same places ("HDR").

Author Response

Typographical errors has been corrected

Thanks

Reviewer 2 Report

The review describes a comprensive overview of studies dealing with MPM therapy. The paper is well written and covers the present situation quite well.

Nevertheless the title is somewhat misleading, as "oncological frontiers" are not discussed, but studies are cited grouped according to  their target mechanisms.   A visionary contribution as promised in the title is not found in the paper.

Author Response

Conclusions has been modifed in order to emphasize future research directions 

Reviewer 3 Report

In general this review gives a clear state of the art overview of available  and possibly emerging systemic therapies for malignant mesothelioma. It is wel written.

In the abstract some corrections should be made:

line 17  : has rapidly emerged          instead of       emerging

line 19 : may open                                "       "         is opening

line 20 : possibly enrich                       "        "         likely

line 20 : and thereby complicate        "        "          but also dramatically complicate

In general in the abstract, targeted drugs are described too optimistic according to their actual positioning. At the best they are promising in their antitumour activity, clinical activity has still to be proven.

The tables give a good overview, but references to the literature listed should be added to the names of the studies.

Many references numbers do not correspond with the right article. For instance number 34 in the text corresponds to the listed article 35     

In the introduction it is stated that Multimodality therapy including surgery is offered to young patients with good Performance status, localized disease and epitheliod histology. This statement contrasts with the content of the Critical Discussion. The role of surgery is better described as limited than cotroversial. Moreover, multimodality regimes have been studied prospectively, but only a few trials were randomized

Author Response

  • Corrections in the abstract and discussion have been done according to indications
  • Mistakes in the references was corrected
  • Reference number has been added in the tables for published studies